# Primary Pulmonary Carcinomas with Spindle and/or Giant Cell Features: A Review with Emphasis in Classification and Pitfalls in Diagnosis

**DOI:** 10.3390/diagnostics13152477

**Published:** 2023-07-26

**Authors:** Cesar A. Moran

**Affiliations:** MD Anderson Cancer Center, University of Texas, Houston, TX 77030, USA; cesarmoran@mdanderson.org

**Keywords:** carcinoma, giant cells, spindle cells, immunohistochemistry

## Abstract

Primary carcinomas of the lung are vastly represented by the conventional types of adenocarcinomas or squamous cell carcinomas. However, there are other types of non-small cell carcinomas that although uncommon represent a meaningful group that often pose a problem not only in diagnosis but also in classification. Spindle cell and/or giant cell carcinomas, although uncommon represent an important group of primary lung carcinomas. Important to highlight is that current criteria are rather ambiguous and likely not up to date, which renders the classification of these tumors somewhat more obscure. In addition, with the daily use of immunohistochemical stains, the classification of these tumors may also pose a different problem in the proper allocation of these tumors. Proper classification is highly important in the selection process that takes place using such material for molecular analysis. The current molecular characteristics of these tumors are limited and lack more in-depth studies and analyses that can provide specific targets for the treatment of patients with these tumors. The current review attempts to highlight the shortcomings in the current classification and definitions of these neoplasms as well as the more current view regarding these tumors when the use of immunohistochemical stains is employed.

## 1. Introduction

Primary carcinomas of the lung are dominated by the conventional types, namely adenocarcinoma and squamous cell carcinoma. In the current practice, with the use of immunohistochemical analysis using markers to either document pneumocytic or squamous differentiation, namely, the use of p40, keratin 5/6, p63, TTF-1, Napsin A, the vast majority of non-small cell carcinoma can be specifically categorized. The percentage of non-small cell carcinoma that do not show specific lineage for either pneumocytic or squamous differentiation is rather limited to no more than 2–3%. However, there is a small percentage of primary malignant neoplasms of the lung that show morphological features that depart from the conventional histologies and that may be composed of spindle cells and/or giant cells. This group of tumors, although well-recognized in the literature, for the most part, it has been coded under different designations in the past [1,2,3,4,5]. Even though some tumors may show an additional component of the conventional non-small cell carcinoma, there are some other tumors that may be exclusively composed of either spindle or giant cells.

In this review, the presence of these components will be highlighted, either in association with a conventional non-small cell carcinoma or when the tumors occur with exclusive features of ‘Sarcomatoid” or giant cell carcinomas. In this context, the use of immunohistochemical stains will also be highlighted to properly triage the specific lineage of these tumors whenever possible.

## 2. Historical Perspective

The occurrence of spindle and/or giant cell components in lung carcinomas or even the unusual occurrence of pure sarcomatoid or giant cell carcinomas, has been described in the literature. However, one of the largest issues has been to determine the percentage of giant or spindle cell components to define a tumor as mixed, predominantly, pure sarcomatoid, or giant cell carcinoma. In that respect, previous publications from the World Health Organization (WHO) [6], provided little light into those definitions. In the 2004 histological classification of lung tumors by the WHO [7], pleomorphic carcinoma, spindle cell carcinoma, giant cell carcinoma, carcinosarcoma, and pulmonary blastoma appear under the heading of “Sarcomatoid Carcinoma. In the description of “pleomorphic carcinoma”, a cut-off of 10% of the malignant spindle or giant cells is provided. In 2015, the WHO classification of lung tumors [8] lumped together “pleomorphic carcinoma, spindle cell, and giant cell carcinoma”, and stated that these tumors should contain at least 10% spindle and/or giant cells or a carcinoma consisting of only spindle and giant cells. It further states that giant cell carcinomas “consist almost entirely of tumor giant cells with no differentiated carcinomatous elements”. In the most recent publication from the WHO [9], pleomorphic carcinoma, pulmonary blastoma, and carcinosarcoma are included under the heading of sarcomatoid carcinomas, and the definition for the 10% cut-off remained. 

At this juncture, it is important to evaluate the rationale behind the cut-off of 10%. How was it determined? How scientifically accurate is the determination of 10%? To shed some light on those questions, it is time to go back to the only publication from where such a percentage was determined. 

In 1994, Fishback et al. [10] reviewed the files of the pulmonary and mediastinal branch of the Armed Forces Institute of Pathology (AFIP) and identified 1128 accessioned cases of carcinoma with spindle or giant cell features and pleomorphic carcinomas of the lung over a period of 20 years (1971–1991). Of those 1128 cases, the authors selected 78 cases that were classified as having components of adenocarcinoma, squamous cell carcinoma, spindle cell carcinoma, giant cell carcinoma, clear cell carcinoma, and small cell carcinoma. The authors stated that to avoid the inclusion of cases with only scattered giant cells, a minimum requirement of 10% of the giant cell population was set. It is highly important to highlight that the material available for the establishment of the 10% cut-off consisted of 57 cases in which a surgical resection was available (wedge resection, lobectomy, pneumonectomy). It is evident that the established 10% was not based on any scientific criteria or special study, but merely to facilitate the inclusion of cases that the authors had already determined. In addition, even if we consider this 10% appropriate as a cut-off, there was no rationale in determining this percentage based on the size of the tumor nor based on the histological sections available for review. Indeed, the author reviewed all histological material available but there are no data to support how many sections of tumor were available for review. Furthermore, in 1994 the development of immunohistochemistry was not nearly as it is today, and that can be easily determined by the basic immunohistochemical studies reported in that study, which essentially was based on pan-keratin, vimentin, and epithelial membrane antigen. Interestingly, the authors stated that histologically, 22% of the 78 cases reported showed exclusively spindle cell components and added that 30 of the 78 cases had a giant cell component in association with the spindle cell component, while only 18 of the 78 cases had only giant cell components. The authors determined that spindle cell carcinoma was present in 60 of the 78 cases reported, while giant cell carcinoma was seen in 48 of the 78 cases. The most common association was spindle cell carcinoma with giant cell component. Other histological types that were also seen in association included adenocarcinoma, squamous cell carcinoma, large cell carcinoma, and small cell carcinoma. The authors suggested the use of the term “pleomorphic carcinoma” for tumors with spindle and/or giant cell carcinoma and added the possibility that these tumors may represent a subtype of large cell carcinoma. 

In retrospect, it is obvious that the 10% cut-off was arbitrarily determined by the authors who could have established a lower or higher percentage and still it would have been without specific and more accurate data. The size of the tumor and the number of tumor sections evaluated likely represent the most accurate way to establish a more scientific cut-off; however, such determination over the last 30 years has remained elusive, mainly in a modern era of immunohistochemistry and molecular techniques, where personalized medicine plays an important role. Even though Fishback et al. [10] provided a concept to unify tumors mainly those with spindle and giant cell components under the designation of “pleomorphic carcinoma”, the authors also left unanswered many other important issues such as what to do with tumors that show a meaningful giant cell component and another non-small cell carcinoma? Should these tumors be called giant cell carcinomas? Non-small cell carcinoma (Adenocarcinoma—squamous cell carcinoma) with giant cell component? According to the authors, they were able to determine that foci of squamous cell carcinoma, adenocarcinoma, or large cell carcinoma can be seen in what they coded as “pleomorphic carcinoma in a range of 8% to 45% but do not specifically address any percentage. Those questions go to the core of tumor classification and likely in the current era of personalized medicine and molecular diagnostics, those specific designations could be evaluated further, as they may play a role in clinical outcomes. Some of these issues have been raised in more recent manuscripts dealing with these specific features. One additional drawback is the “simplistic” designation of giant cells, which since the series presented by Fishback [10] has remained intact as defined by the authors as “cells with abundant cytoplasm containing multiple nuclei or a single large pleomorphic nucleus”. 

## 3. Analysis of the Literature

By far most non-small cell carcinomas of the lung are represented by conventional adenocarcinoma and squamous cell carcinoma. The use of immunohistochemical studies has also contributed to further characterize tumors that in the past would have been coded under the terminology of “large cell carcinoma [11]”. Therefore, cases of large cell carcinoma represent a minority with likely less than 1–2%, as immunohistochemistry and molecular diagnostics likely play an important role in properly designating cases that on histology alone do not show specific differentiation. A similar analogy can be drawn with primary malignant tumors of the lung that may show either spindle cell component, giant cell component, or a mixture of these components with or without the association of the conventional histologies.

In 2017, Weissferdt et al. [12] evaluated by immunohistochemical means 86 cases of spindle and pleomorphic carcinomas following somewhat the same criteria already presented by Fishback et al. [10]. The authors of this immunohistochemical study now using more up-to-date immunohistochemical analysis with antibodies for pneumocytic and squamous differentiation (TTF-1, Napsin A, keratin 5/6 and p40) encountered that 44% of primary tumors initially classified as “sarcomatoid” could be re-classified as Adenocarcinomas, while 14% could be re-classified as squamous cell carcinoma. It is important to highlight that in 36 of the 86 cases evaluated in which the tumors could be re-classified as adenocarcinomas, or, in 12 of 86 cases for squamous cell carcinoma, the positive staining for TTF-1 and/or Napsin A, or keratin 5/6 and/or p40, was in the spindle/giant cell component of the tumor. Following the experience with the immunohistochemical analysis of 86 cases of spindle cell and pleomorphic (“sarcomaotid”) carcinomas of the lung, Weissferdt et al. [13] presented a novel perspective in tumor classification with the goal of proper triaging of these cases and offering patients the possibility of more targeted treatment options. The authors proposed a classification based on histology and immunohistochemical profile of those tumors, creating a specific designation for those tumors as follows: Sarcomatoid carcinoma + Conventional Adenocarcinoma:
Pneumocytic markers positive in the spindle cell component (TTF-1 and/or Napsin A):Sarcomatoid Adenocarcinoma
b.Pneumocytic markers negative (TTF-1 and/or Napsin A):Dedifferentiated AdenocarcinomaSarcomatoid carcinoma + Conventional Squamous cell carcinoma:Squamous markers positive in the spindle cell component (keratin 5/6 and/or p40)Sarcomatoid squamous cell carcinomaSquamous markers negative (keratin 5/6 and/or p40):Dedifferentiated squamous cell carcinoma.Sarcomatoid carcinoma + Carcinoma without morphological differentiation towards Adenocarcinoma or squamous cell carcinoma:Positive pneumocytic markers = Sarcomaotid AdenocarcinomaPositive squamous markers = Sarcomatoid squamous cell carcinomaNegative penumocytic or squamous markers = Sarcomatoid Large cell carcinoma.

Using these specific criteria, the authors were able to re-classify 42% of adenocarcinomas, sarcomatoid type; 15% of squamous cell carcinomas sarcomatoid type, 15% as dedifferentiated adenocarcinomas, and 28% as sarcomatoid large cell carcinomas. In addition, the authors argue that such triaging of cases not only provides a more accurate pathological designation for these tumors but also provides more accurate information to oncologists for the possible selection of treatment. Such a claim has been made also by other authors who also concur in more accurate profiling for tumors that depart from the conventional histologies as those tumors may also show similar molecular profiling as those with more conventional histology [14,15,16,17,18,19].

Although the emphasis in most of the reports has been on the presence of the “sarcomatoid” component, there is also another component that is often encountered—the giant cell component. Such a component although known and reported in the literature [20,21,22,23,24], has also been controversial regarding the type of giant cells present. In addition, in most of the documented cases in which the presence of giant cells has been extensive, there is little information regarding the type of giant cells, even though it has been stated that those giant cells are epithelial in origin [25,26,27]. In some cases, due to the similarity of the giant cells with those present in other tumors such as choriocarcinomas (syncytiotrophoblastic cells) plus the expression of human chorionic gonadotrophin in the giant cells, the designation for these tumors has been that of primary choriocarcinoma of the lung [28,29,30,31,32]. One important aspect that is important to highlight is that over the years these tumors appear to be classified by the WHO under headings that are likely incorrect—under large cell carcinoma and in the most recent publication under “Sarcomatoid” carcinoma. 

More recently to bring more clarity to the subject of giant cell carcinomas, a study of seven cases was presented in which more state-of-the-art immunohistochemistry was performed with more specific antibodies [33]. The authors documented cases with extensive presence of giant cells in which the tumors did not show any morphological evidence of differentiation towards any of the known non-small cell carcinomas (adenocarcinoma or squamous cell carcinoma). In addition, none of the patients had any increase in serum level of human chorionic gonadotrophin. By morphology and immunohistochemistry, the authors were able to separate two different types of giant cells: (1) syncytiotrophoblast-like giant cells were characterized by positive staining for human chorionic gonadotrophin but negative staining for pneumocytic and/or squamous markers (TTF-1, Napsin, and p40), and (2) emperipoletic/null type giant cell characterized by positive staining for keratin but negative staining for human chorionic gonadotrophin, and pneumocytic and squamous markers. In addition, to these two different types of giant cells in lung carcinomas, Lindholm et al. [34] reported three cases that the authors designated as osteoclast-like giant cell-rich carcinomas of the lung. These giant cells appear to show positive staining for CD-68, cathepsin K, and histone H3 and negative for pneumocytic and squamous markers. In two cases there was a sarcomatoid component and in one case adenocarcinoma component. In addition, the authors documented that in the molecular analysis ALK, BRAF, EGFR, ROS1, RET, and MET were negative. 

## 4. Clinical Features

In the largest series of these tumors, there does not appear to be a predominant gender although men appear to be slightly more affected than women. The average age for the appearance of this tumor is about 63 years. The symptomatology of these patients will vary depending on the location and the size of the tumor. Patients with tumors in a central location will show symptoms of obstruction such as dyspnea, cough, and shortness of breath, while patients with peripheral tumors are likely to present with chest pain and shortness of breath. 

## 5. Pathological Features

### 5.1. Biopsy Interpretation

One important shortcoming in the interpretation of spindle and/or giant cell carcinomas is when the only tissue available for interpretation is a small biopsy specimen. Often pathologists are faced with the interpretation of small fragments of tissue from patients with a large pulmonary mass but in which the patient may be in the advanced stages of the disease requiring medical intervention rather than surgical resection. These cases pose a significant problem not in the interpretation of malignancy but rather in the interpretation of the specific subtype of carcinoma that could be further analyzed by advanced methods such as molecular techniques. Classifying these tumors as “Sarcomatoid” may be to some extent misleading as one is evaluating only a minor percentage of the tumor, even though that fragment of tissue may sow the morphological features of a spindle cell neoplasm. In such cases, two additional paths of care should be followed: (1) the use of more current immunohistochemical methods to properly determine whether the tumor has squamous or pneumocytic differentiation. Having the benefit of more specific immunohistochemical stains (pneumocytic and squamous markers), the tumor may be assigned to a specific category stating that the tumor has a spindle or giant cell morphology; (2) in cases in which the pneumocytic and squamous markers are negative and only keratin is positive, the appropriate interpretation in a small fragment of tissue would be that of non-small cell carcinoma with spindle and/or giant cell features that may represent pleomorphic carcinoma. Nevertheless, the definitive classification of such tumors should be performed only after a surgical resection takes place so that more sampling is available for interpretation and proper immunohistochemical stains are performed. 

In terms of molecular profiling, if the material available for such analysis is the small biopsy, it should be carefully stated that even though the morphology is that of a “sarcomatoid” carcinoma, more definitive classification should not be based on this small fragment of tissue but after a surgical resection becomes available. If the results of the molecular profiling results are those that may be seen in adenocarcinomas or squamous cell carcinoma, then such tumor should be allocated in that subclassification.

### 5.2. Macroscopic Features

Tumors that histologically show spindle and/or giant cells cannot be separated on macroscopic grounds from other types of non-small cell lung carcinomas. The tumors can be centrally or peripherally located. The tumor size has been described as ranging from 2 to more than 10 cm in greatest diameter, with or without areas of necrosis or hemorrhage. When the tumors are not necrotic, the color can vary from white to gray and may have soft or mucoid consistency [10,13,33]. The only tumor that appears to show a different color is the one that is rich in osteoclast giant cells, which shows a reddish color [34].

### 5.3. Microscopic Features

The different histopathological features and the respective immunohistochemical analysis is presented in Table 1.

*Sarcomatoid carcinomas:* These tumors show a tightly packed spindle cell proliferation composed of slender cells with fusiform nuclei and inconspicuous nucleoli, replacing normal lung parenchyma. The tumors are well delimited but not encapsulated (Figure 1). Cellular atypia is variable and may show areas of mild to moderate to marked atypia. Mitotic figures also vary and may be inconspicuous or may be evident with the presence of atypical mitotic figures (Figure 2A,B). In high-grade tumors, the presence of necrosis and hemorrhage is prominent and is mixed with the neoplastic component. Important to recognize is that sarcomatoid carcinomas may be associated with areas of otherwise conventional non-small cell carcinoma such as adenocarcinoma or squamous cell carcinoma (Figure 3A,B). In addition, sarcomatoid carcinoma may also show the presence of bizarre giant cells admixed with the spindle cell component (pleomorphic carcinoma) (Figure 4).

*Giant Cell Carcinomas:* These tumors may show predominantly a neoplastic cellular proliferation composed exclusively of multinucleated giant cells or a predominantly giant cell carcinoma (Figure 5A,B) or associated with a conventional non-small cell carcinoma like adenocarcinoma or squamous cell carcinoma. The giant cell carcinoma may show giant cells of the syncytiotrophoblastic, osteoclastic, or null cell type. The giant cell carcinomas of the null cell type characteristically show a prominent inflammatory background and giant cells engulfing inflammatory cells (emperipolesis) (Figure 6A,B). The tumors composed of osteoclast-like giant cells show giant cells like those described in bone tumors (Figure 7A,B).

#### Immunohistochemical Features

The use of conventional pneumocytic and squamous markers such as TTF-1, Napsin A, p40, p63, and keratin 5/6 are commonly used in the evaluation of non-small cell carcinomas. These markers also play an important role in the evaluation of the spindle cell component as it has been demonstrated that the spindle cells may show positive staining for either pneumocytic or squamous markers, which will provide a more accurate classification of these tumors (Figure 8A–D). On the other hand, the use of other markers such as human chorionic gonadotrophin, cytokeratin, CD68, cathepsin, and histone H3 may provide important information in the type of giant cells present, thus a more accurate classification of these tumors.

The larger single most important issue regarding immunohistochemical analysis is the type of tissue for evaluation. In daily practice that may pose significant challenges as the only tissue may be a small biopsy, which inevitably will have limitations, mainly if the tissue is negative for all specific markers (squamous or pneumocytic). In such cases, the term “sarcomatoid” carcinoma may become a default diagnosis. However, every effort should be made to correlate that biopsy with a possible surgical resection in which more tissue becomes available. Additionally, important to mention is that diagnostic surgical pathologists are limited to the number of possible stains that may be available for proper classification –TTF-1, Napsin A, p63, p40, keratin 5/6 are among the most commonly used and recommended for diagnostic purposes, but it is also known that some of those stains may also show cross-reactivity, e.g., p63 commonly used as squamous marker is positive in about 25% of adenocarcinomas. However, in such cases at least one can state that there is some immunohistochemical evidence of differentiation towards what the morphology may dictate. Also the current definition of these tumors at the end may play a role in how tumors are classified.

### 5.4. Molecular Features

The evaluation of tumors composed exclusively of spindle or giant cells is still a work in progress. One important aspect is that these tumors are not very common in comparison to the conventional non-small cell carcinomas. In addition, when these tumors are evaluated, usually it is the non-small cell component that is associated with the spindle cell or giant cell carcinoma. Therefore, there is existing bias in their evaluation. However, this issue has been highlighted by some authors about the need to properly analyze these types of tumors [35,36]. Currently, some studies on sarcomatoid carcinoma have been performed [37,38,39,40,41,42] showing some variations in the molecular analysis such as MET exon 14 skipping mutations. However, the issue of giant cell carcinomas remains unknown as such a component has eluded a more comprehensive analysis.

In addition, other molecular features that have been encountered in sarcomatoid carcinomas include higher prevalence of TP53, KRAS, PIK3CA, MET, NOTCH, STK11, and RB1. However, the correlation that has not taken place is whether those tumors studied could have been classified as spindle cell squamous cell carcinomas, spindle cell adenocarcinomas, or just spindle large cell carcinomas. Such analysis could explain the results in some of these cases and in addition provide important information to oncologists for the possible treatment of such patients. One additional issue to highlight is whether the material available was part of a small biopsy; whether it came from a resected specimen, and whether specific immunohistochemical markers were performed and the results of them. These are important issues to address so that proper classification is provided rather than the general term of “sarcomatoid” carcinoma. We should be aware of the limitations that exist in biopsy interpretation. However, in such cases, if a surgical resection takes place, it may be possible to correlate the biopsy, the resected material, and the molecular features that may have become available in case the biopsy was the tissue that was used for molecular testing. Having that correlation may provide important information that can be used for prospective studies and analysis. Currently, due to the definition provided for these tumors, it is likely that molecular studies may also show some incomplete information. 

## 6. Differential Diagnosis

The differential diagnosis of spindle cell carcinoma of the lung can be wide as the tumor may mimic spindle cell sarcomas, either primary sarcoma of the lung or metastatic sarcoma from a soft tissue primary. When the tumors are composed of giant cells the possibility of metastatic sarcoma with giant cells needs to be explored even though primary giant cell tumors of the lung have been described [43]. Therefore, the use of a wider panel of immunohistochemical stains becomes important in the proper classification of these tumors. In addition, the clinical history of an extra-thoracic neoplasm must be properly excluded by clinical means.

However, it is important to highlight that the interpretation of a biopsy specimen may pose limitations in interpretation. Even in cases in which the small biopsy may show a sarcomatoid neoplasm, the larger issue is whether that small fragment of tissue is truly representative of the intrapulmonary mass or whether the spindle cell component represents only a minor component of the tumor in question. One additional problem would be in cases in which the tumor is largely composed of a mesenchymal component, which may possibly lead to the interpretation of “carcinosarcoma”. It is advisable that the final interpretation of a sarcomatoid and/or giant cell carcinoma be conducted after a surgical resection becomes available and proper sampling and immunohistochemical stains are performed.

## 7. Summary

Although the knowledge of the most conventional types of non-small cell carcinoma (adenocarcinoma, squamous cell carcinoma) has advanced dramatically over the last 10–20 years, it is also evident that unusual non-small cell carcinomas such as spindle cell carcinoma and giant cell carcinoma not only need a better histological definition but also more advance analysis using molecular techniques. The current classification of these tumors using an arbitrary 10% is not appropriate as a conventional adenocarcinoma or squamous cell carcinoma with “the 10%” could automatically be classified as sarcomatoid or giant cell carcinoma. The criteria for separation of these tumors require a deeper analysis not only considering the size of the tumor but also the use of proper immunohistochemical analysis. It has been already demonstrated that numerous cases that are otherwise classified as “sarcomatoid” or “pleomorphic” carcinomas may fall into a more specific classification if proper immunohistochemical stains are employed, and depending on those results, these tumors should be allocated to one of the most specific categories. In that way, more advanced techniques such as molecular analysis could provide a better guide for the identification of new targets or molecular alterations. In addition, the presence of giant cells either as a component of a conventional non-small cell carcinoma or a tumor composed of only giant cells remains a subject that deserves better understanding.

One important and unavoidable issue is the fact that often only small biopsies are available for additional studies, which may preclude specific classification using immunohistochemical methods. Those small biopsies are also often used for molecular analysis which may provide important but skewed information regarding the tumor in question. For instance, the molecular findings of spindle cell proliferation may in fact be a small component of a morphologically adenocarcinoma or squamous cell carcinoma. On the other hand, such spindle cell components may in a resected specimen show positive staining for pneumocytic or squamous markers rendering the subclassification more towards those tumors. The recommendation would be that if a small biopsy showing spindle cell carcinoma is used for immunohistochemical stains and/or molecular analysis, a further correlation should take place when and if a surgical resection is performed and place it in the context of the molecular results. This methodology will likely increase the threshold for a specific classification and will lead the path towards future prospective studies. In addition, it will provide the necessary information that oncologists need to choose specific lines of treatment.

## Figures and Tables

**Figure 1 diagnostics-13-02477-f001:**
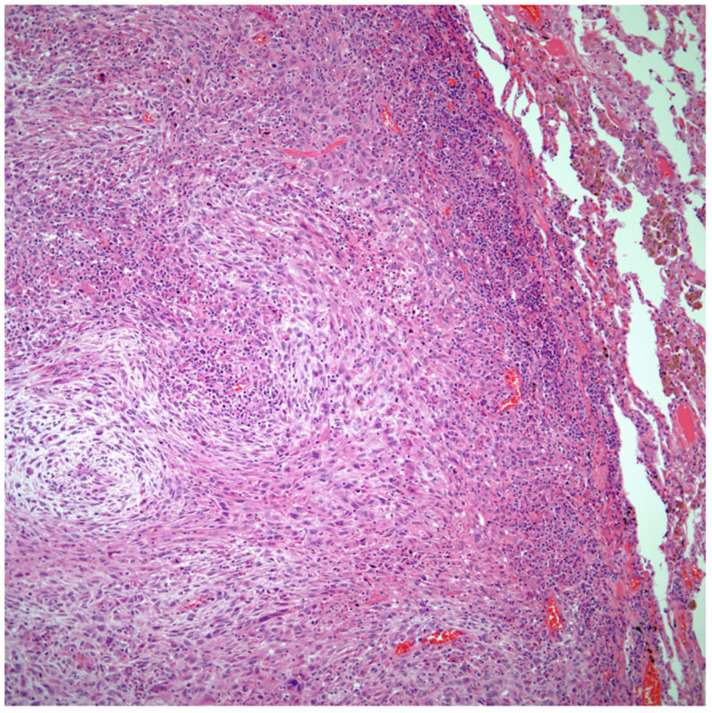
Sarcomatoid carcinoma of the lung showing a well circumscribed tumor replacing lung parenchyma.

**Figure 2 diagnostics-13-02477-f002:**
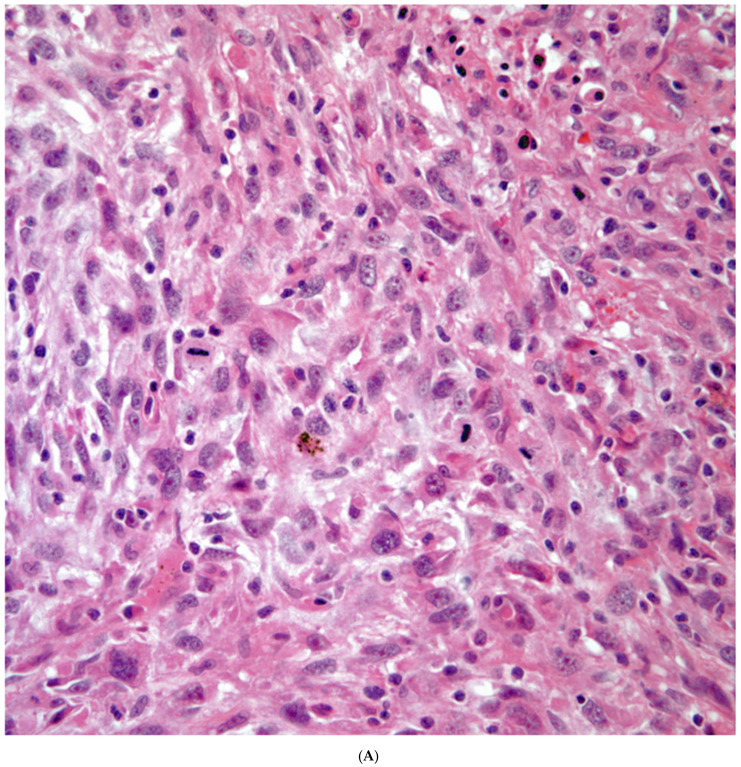
(**A**) Atypia and mitotic activity. (**B**) Neoplastic spindle cell proliferation.

**Figure 3 diagnostics-13-02477-f003:**
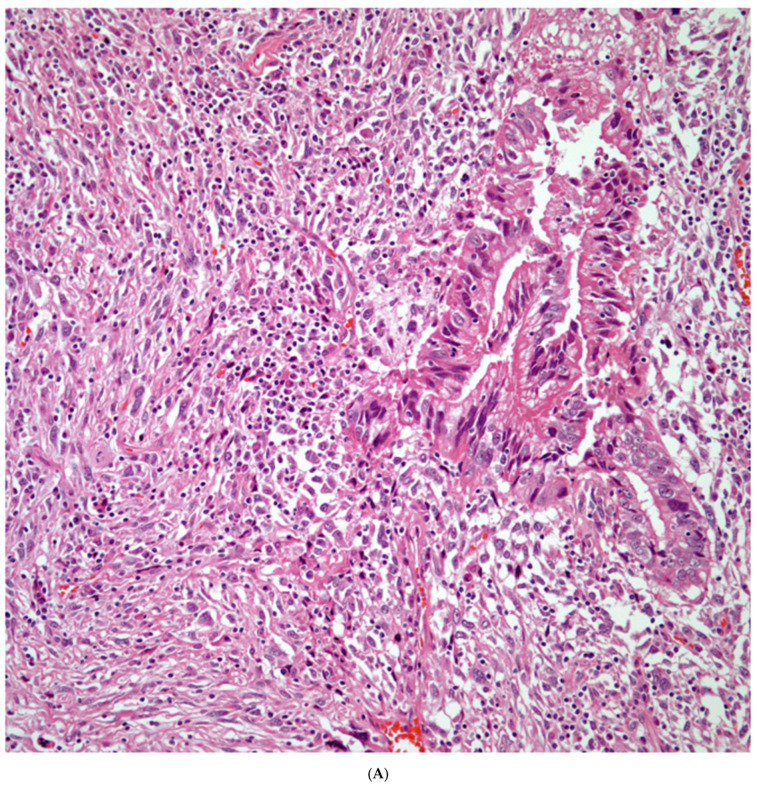
(**A**) Sarcomatoid carcinoma associated with areas of conventional adenocarcinoma; (**B**) Sarcomatoid carcinoma associated with areas of squamous carcinoma.

**Figure 4 diagnostics-13-02477-f004:**
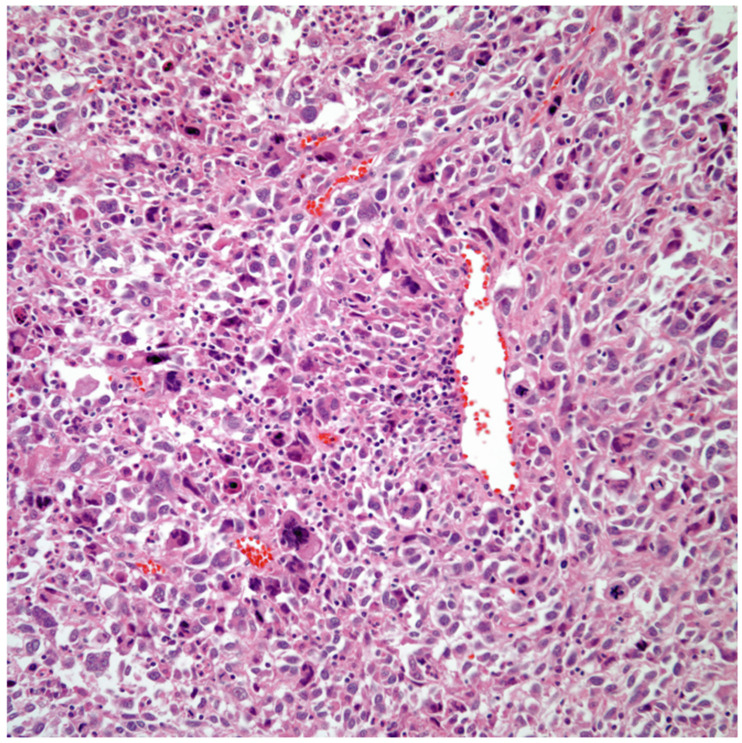
Sarcomatoid carcinoma with giant cell component (pleomorphic carcinoma).

**Figure 5 diagnostics-13-02477-f005:**
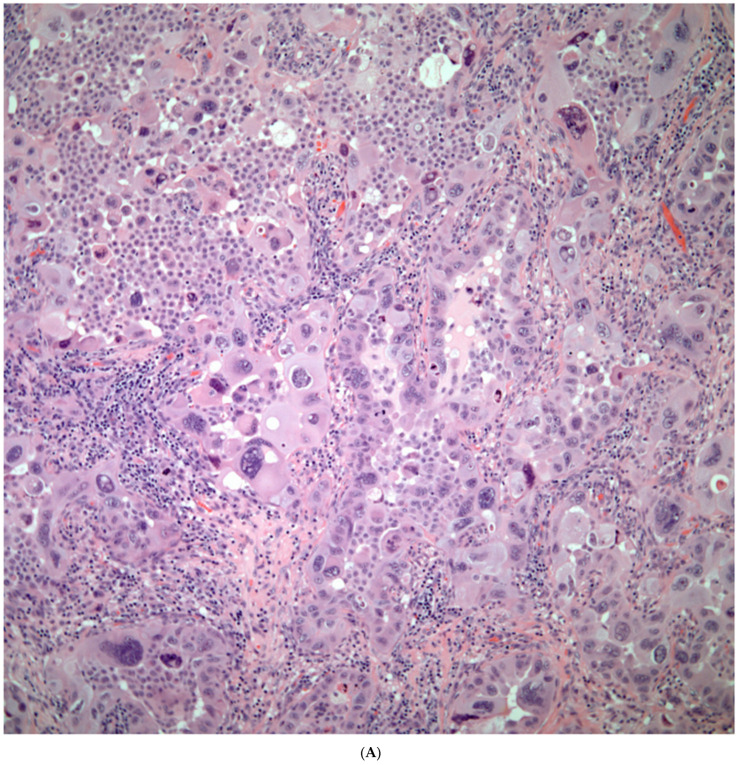
(**A**) Predominantly giant cell carcinoma; (**B**) Marked atypia and numerous multinucleated malignant giant cells.

**Figure 6 diagnostics-13-02477-f006:**
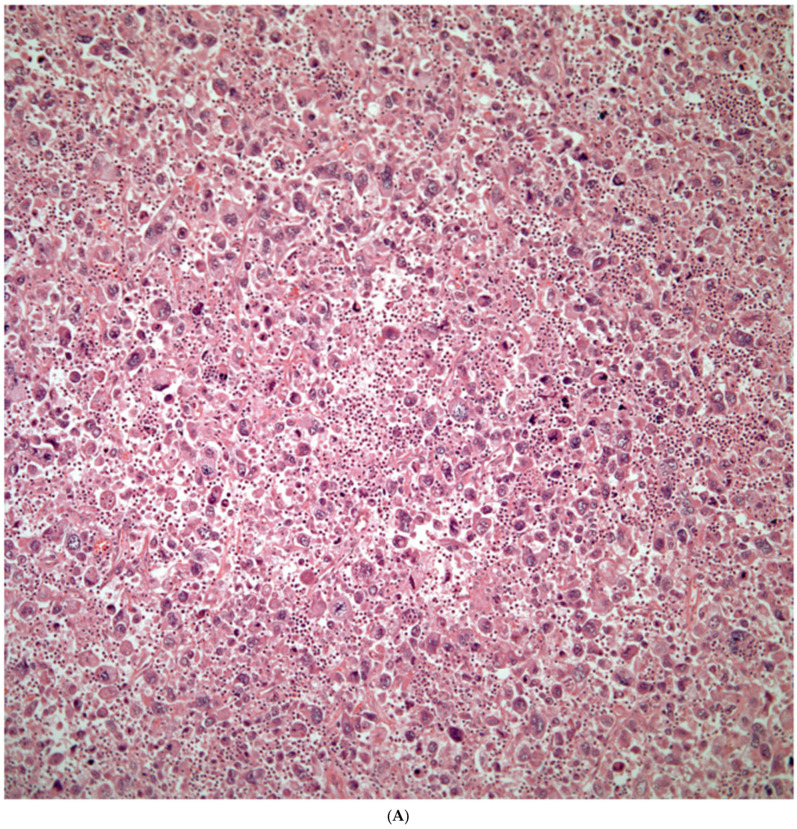
(**A**) Giant cell carcinoma, null cell type, note the inflammatory background; (**B**) Malignant giant cells with inflammatory cells and focal emperipolesis.

**Figure 7 diagnostics-13-02477-f007:**
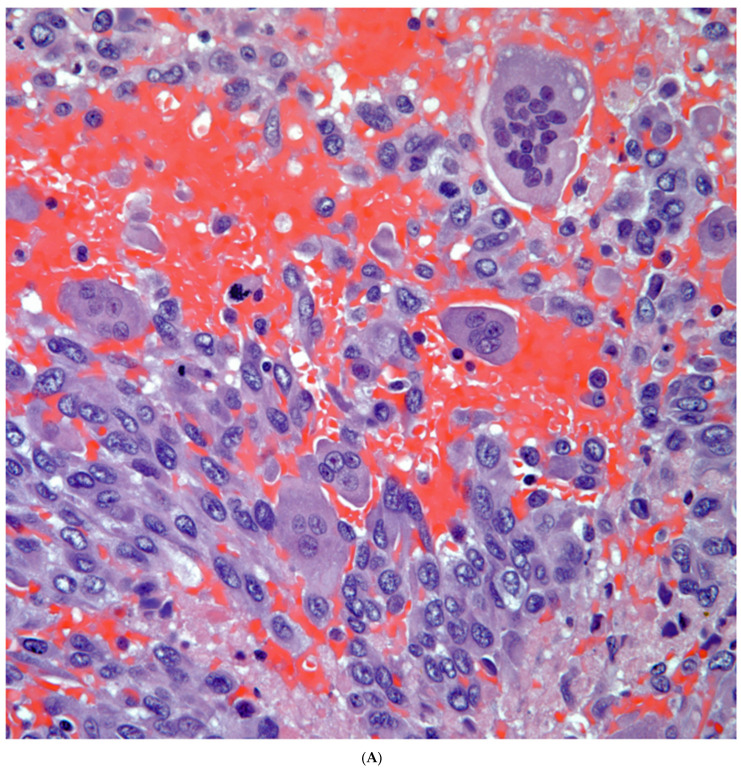
(**A**) Carcinoma associated with osteoclast giant cells; (**B**) Osteoclast giant cells like those in bone tumors.

**Figure 8 diagnostics-13-02477-f008:**
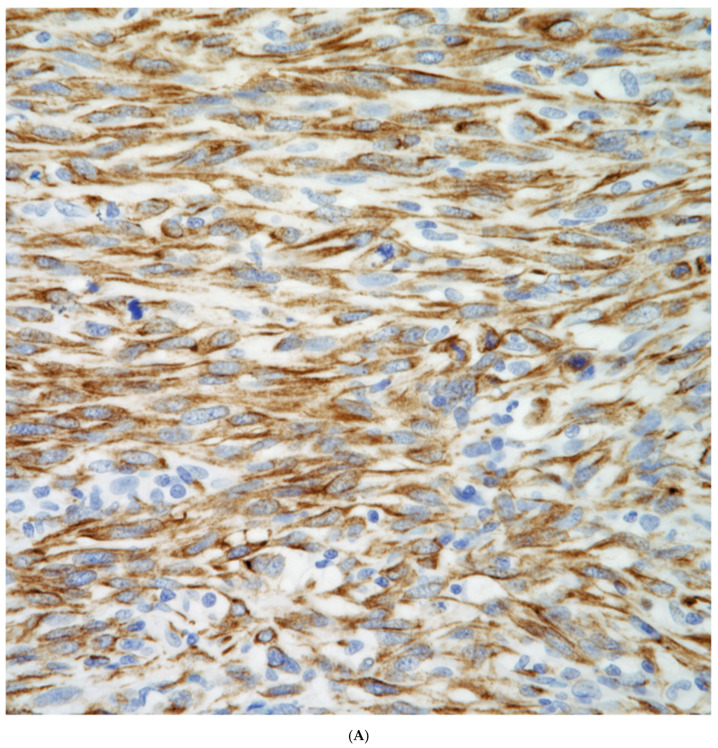
(**A**) Keratin positive in a sarcomatoid carcinoma; (**B**) p40 positive in a sarcomatoid squamous cell carcinoma, (**C**) TTF-1 positive in a pleomorphic carcinoma; (**D**) HCG positive in multinucleated giant cells.

**Table 1 diagnostics-13-02477-t001:** Combination of morphological features, immunohistochemical features with suggested interpretion.

Tumor Characteristics	Immunohistochemistry	Interpretation
Pure Spindle cell morphology	Keratin positive spindle cells	
	Negative TTF-1 and Napsin	
	Negative p40/p63/keratin 5/6	Sarcomatoid carcinoma
Pure spindle cell morphology	Positive TTF-1/Napsin	Sarcomatoid Adenocarcinoma
Pure spindle cell morphology	positive p40/p63/keratin 5/6	Sarcomatoid squamous cell carcinoma
Mixed Spindle and giant cells	keratin positive spindle cells	Pleomorphic carcinoma
Mixed spindle with areas of adenocarcinoma	TTF-1/Napsin positive spindle cells	Sarcomatoid Adenocarcinoma
Mixed spindle cells with areas of adenocarcinoma	TTF-1/Napsin negative in spindle cells	Dedifferentiated Adenocarcinoma
Mixed spindle cell with areas of squamous carcinoma	p40/p63/keratin 5/6 positive spindle cells	Sarcomatoid squamous cell carcinoma
Mixed spindle cells with areas of squamous carcinoma	Negative p40/p63/keratin 5/6 in spindle cells	Dedifferentiated Squamous cell carcinoma
Pure Giant cell morphology	giant cells positive for keratin only	Giant cell carcinoma, Null cell type
Pure Giant cell morphology	giant cells positive for HCG	Giant cell carcinoma with syncytiotrophoblast-like
		Cells
Pure giant cells with areas of adenocarcinoma	giant cells positive for keratin and TTF-1/Napsin	Giant cell adenocarcinoma
Pure giant cell with areas of adenocarcinoma		
Or squamous carcinoma	giant cells positive for CD68/cathepsin/histone H3	Adenocarcinoma or squamous cell carcinoma with
		Osteoclast giant cell component.

## Data Availability

Not applicable.

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
