# Peer review of "Primary Pulmonary Carcinomas with Spindle and/or Giant Cell Features: A Review with Emphasis in Classification and Pitfalls in Diagnosis"

_diagnostics, 2023, doi:10.3390/diagnostics13152477_

Round 1

Reviewer 1 Report

The author provided a synopsis of the current diagnostic outlook on two infrequent subtypes of primary lung carcinomas, spindle and giant cell carcinomas. The following suggestions might enhance the quality of this review:

1. There's a need for a more comprehensive discussion on the molecular and genetic characteristics, including but not limited to, alterations in chromosomes, gene mutations, and polygenic fusions.

2. Generally speaking, the author could strengthen the conclusions by citing more related reference studies, especially in the immunohistochemical features discussion at line 281.

3. The statement on line 15, 'Spindle and giant cell carcinomas, although uncommon primary lung carcinomas, are well known to occur,' requires rephrasing.

Author Response

The recommendations of the reviewers have been included in the revision and have been highlighted in yellow.  However, it is important to highlight also of the current limitations on sarcomatoid carcinomas of the lung and the lack of larger studies.  Similarly, the limitations on specific immunohistochemical markers has been noted.

In general, I agree with he reviewers and have added more text to the manuscript keeping the needed balance to it.

Reviewer 2 Report

Pulmonary carcinoma with spindle and/or giant cell features is an uncommon lung non-small cell carcinoma, yet it often causes diagnostic confusion, misclassification and possible under treatment. Correct classification and pathology reporting of the special carcinoma can facilitate necessary molecular testing and provide oncologists with comprehensible information for patient’s treatment/management. Your review has addressed this special type of carcinoma for the pathology diagnosis and possible implication for molecular testing and patient’s treatment.

I totally agree with you that the >=10% criteria of spindle/giant cell component for its diagnosis are arbitrarily and is too low. Immunohistochemistry stains should be used to further classify it into adenocarcinoma or squamous cell carcinoma with spindle cell/giant cell component.  Pathologists should try our best to classify it into either adenocarcinoma or squamous cell carcinoma.  

If the tumor is negative for markers of adenocarcinoma and squamous cell carcinoma, then pleomorphic carcinoma can be used for resection specimen with a comment that it is still a variant of non-small cell carcinoma, and  eligible for molecular testing. Such comment can eliminate the oncologist’s confusion about the pathology diagnosis. 

There are a few minor things I would like to mention. 

1.    Line 228, Table 1.  I do not see the table in the downloaded manuscript.

2.    Line 243, such should be such as.

3.    As a subspecialized pathologist myself  working in a large cancer center, I infrequently receive lung biopsy specimen diagnosed as “pleomorphic carcinoma” by community pathologists. It is possible that you write several sentences in your review to emphase  that “pleomorphic carcinoma should not be diagnosed on biopsy specimen”  and such biopsy specimen maybe called: “poorly differentiated non-small cell carcinoma with spindle cell/giant cell features or differentiation” with an explanation that it could represent a pleomorphic carcinoma, but definitive classification  pending resection specimen. 

Author Response

I agree with reviewer regarding the limitations with biopsy specimens in these cases.  Therefore, I added one section on biopsy interpretation following the suggestion provided by the reviewer.

Also have corrected the type.

The table is embedded in the manuscript.